

# Comparative *in-vivo* bond failure rate of orthodontic brackets when bracket base is treated with micro-abrasive blasting *vs.* acid etching: eighteen month randomized control trial and scanning electron microscope study

Owais Khalid Durrani[1], Ulfat Bashir Raja[2], Farooq Ahmad Chaudhary[1], Umar Hamid[3], Muhammad Qasim Javed[4], Sundus Atique[5] and Syed Rashid Habib[6]

[1] School of Dentistry, Shaheed Zulfiqar Ali Bhutto Medical University, Islamabad, Pakistan
[2] Department of Orthodontics, Islamic International Dental College, Riphah International University, Islamabad, Pakistan
[3] Department of Oral Surgery, King's College London Hospital, London, United Kingdom
[4] Department of Operative Dentistry, Islamic International Dental College, Riphah International University, Islamabad, Pakistan
[5] College of Dental Medicine, Qatar University, Doha, Qatar
[6] Department of Prosthetic Dental Sciences, College of Dentistry, King Saud University, Riyadh, Saudi Arabia

Corresponding authors
Owais Khalid Durrani,
drowais@szabmu.edu.pk
Muhammad Qasim Javed,
qasim_javed83@yahoo.com

## ABSTRACT

**Background:** The aim of this study was threefold. Firstly, it aimed to introduce and detail a novel method for chemically etching the bases of stainless-steel orthodontic brackets. Secondly, the study sought to investigate the structural alterations within the brackets' microstructure following chemical etching compared to those with sandblasted bases, using electron microscopy analysis. Lastly, the study aimed to evaluate and compare the long-term durability and survivability of orthodontic brackets with chemically etched bases *versus* those with sandblasted bases, both bonded using the conventional acid etch technique with Transbond XT adhesive, over an 18-month follow-up period.

**Methods:** The study was a randomized clinical control trial with triple blinding and split-mouth study design and consisted of two groups. The brackets in the sandblasted group were prepared by sandblasting the intaglio surface of the base of the bracket with 50 μm $SiO_2$ particles. Hydrofluoric acid was used to roughen the base in the acid-etched group. The bases of the brackets were viewed under an electron microscope to analyze the topographical changes.

**Results:** A total of 5,803 brackets (3,006 acid-etch, 2,797 sandblasted) in 310 patients were bonded, in a split-mouth design by the same operator. The patients were followed for 18 months. The failure rate of 2.59% and 2.7% was noted in an acid-etched and sandblasted group, respectively. There was a close approximation of curves in the Kaplan-Meier plot, and the survival distribution of the two groups in the log-rank (Mantel-Cox) test was insignificant; x2 = 0.062 (*P* value = 0.804).

**Conclusion:** Acid etching if the bases of the brackets can be used as an alternative to sandblasting furthermore acid etching can be performed on the chair side.

# INTRODUCTION

The leap in adhesion technology came with the landmark articles of Buonocore and Newman proposing the idea of acid etching and attaching orthodontic brackets to the tooth surface with acrylic resin (*Buonocore, 1955*; *Newman, 1965*; *Newman, Snyder & Wilson, 1968*). The design of the bracket base is crucial for ensuring optimal adhesion to the tooth surface.

Bracket bases can feature designs such as steel mesh, engravings, sandblasted surfaces, or combinations thereof. Failure in bracket bonding during orthodontic treatment can result in treatment delays, patient discomfort, and increased costs (*Fazal et al., 2023*). Numerous studies have investigated the factors influencing bracket bond strength, identifying enamel surface preparation techniques, adhesive systems, and bracket base properties as critical determinants (*Farahani et al., 2016*; *Izquierdo et al., 2020*). Given its importance, the bracket base continues to be a focal point of research (*Lugato et al., 2009*; *Sharma-Sayal et al., 2003*; *Shyagali et al., 2015*; *Izquierdo et al., 2020*). Research by *MacColl et al. (1998)* and *Sharma et al. (2013)* demonstrated that sandblasted bracket bases of a metal brackets exhibit superior bond strength compared to non-sandblasted bases (*Chaudhary et al., 2019*; *MacColl et al., 1998*; *Sharma et al., 2013*).

Manufacturers often micro-etch or sandblast bracket bases to enhance surface area and improve composite adhesive interlocking. This process typically involves sandblasting with 50–90 µm aluminum oxide or silicon dioxide particles, which, while effective, is labor-intensive and time-consuming. Although hydrofluoric acid is commonly used in the glass industry and for creating glass artwork, it poses significant risks, including skin damage at concentrations below 20% (*Kirkpatrick, Enion & Burd, 1995*; *McKee et al., 2014*). However, in our study, we observed no adverse effects when using hydrofluoric acid at a 10% concentration. The objectives of this randomized controlled trial were threefold: (1) to describe a novel method for chemically etching the bases of stainless-steel orthodontic brackets, (2) to utilize electron microscopy to observe and compare the microstructural changes in orthodontic brackets with sandblasted bases *versus* chemically etched bases, and (3) to evaluate and compare the *in vivo* survivability of orthodontic brackets with chemically etched bases *versus* sandblasted bases when bonded with Transbond XT using the conventional acid etch technique, over an 18-month follow-up period. This study was guided by the hypothesis that chemically etched bracket bases would exhibit a lower failure rate in the oral cavity over the specified period compared to sandblasted bases.

## METHODS

This split-mouth study was designed as a randomized controlled clinical trial with a triple-blind methodology. The CONSORT flow chart, detailing the progression of participants through each stage of the trial, is presented in Fig. 1.

Written consent from the patients and approval from the Ethical Committee of Islamic International Dental Hospital, Riphah International University, were obtained prior to the commencement of this study (IIDC/IRC/06/06/2020). The study was registered with the U.S. National Library of Medicine at clinicaltrials.gov (NCT04456114), and a registry number was issued on June 27, 2020. A total of 310 patients were recruited for this randomized controlled trial. Participant recruitment commenced on June 1, 2018, and was completed on January 31, 2020. The registration with the clinical trial registry was delayed due to technical and administrative issues.

The inclusion criteria for this study were as follows: patients with permanent dentition who were planned to undergo a minimum of 18 months of orthodontic therapy. The bonding of the brackets was performed using the conventional acid etch technique with 37% phosphoric acid etching gel (Scotchbond Universal; 3M, Monrovia, CA, USA), a light-cured bonding agent, and composite (Transbond XT, 3M, Monrovia, USA). The light curing unit (Elipar S10; 3M) was utilized, with the light intensity (1,000 mW/cm²) verified using a light intensity meter (Woodpecker, Guilin, China) before curing the brackets. Consistency was maintained throughout the study by ensuring the same dental chair, the same dental assistant, and the same orthodontist conducted the bonding procedures and followed up with the patients until the end of the trial.

The exclusion criteria for this study were as follows: patients with gross deep bites or crossbites that would affect bracket positioning; any enamel abnormalities, cavitation, or restorations, including crowns on the buccal surface; patients who had previously been treated with fixed orthodontic appliances; patients who had undergone dental bleaching prior to the commencement of their orthodontic treatment; and patients for whom a rapid expander or a fixed functional appliance was planned.

A metal 80-gauge mesh bracket (Orthocare, Saltaire, UK) was chosen for the study. The bracket base was not sandblasted by the manufacturer. The brackets were divided into two groups:

a. **Sandblast:** A dental sandblaster (SandStorm 2; Vaniman Manufacturing Co., Fallbrook, CA, USA) was used to roughen the intaglio surface of the bracket base with 50 μm spherical silicon dioxide particles. The bases were sandblasted for 3 s, maintaining the tip approximately 5 mm away from the bracket base to achieve an even matte surface. The air pressure of the sandblasting machine was kept at 2.5 bars and was constantly monitored for consistency. The brackets were then bathed for 15 min in an ultrasonic instrument cleaner with acetone solution to remove any residual silicon dioxide particles and/or oil droplets that may have embedded on the surface. The brackets were subsequently washed with water, dried, and stored.

b. **Acid Etch:** The bases of the acid-etched group were prepared by applying hydrofluoric acid (HF) (10%) (Merck, Darmstadt, Germany) with a disposable bonding applicator
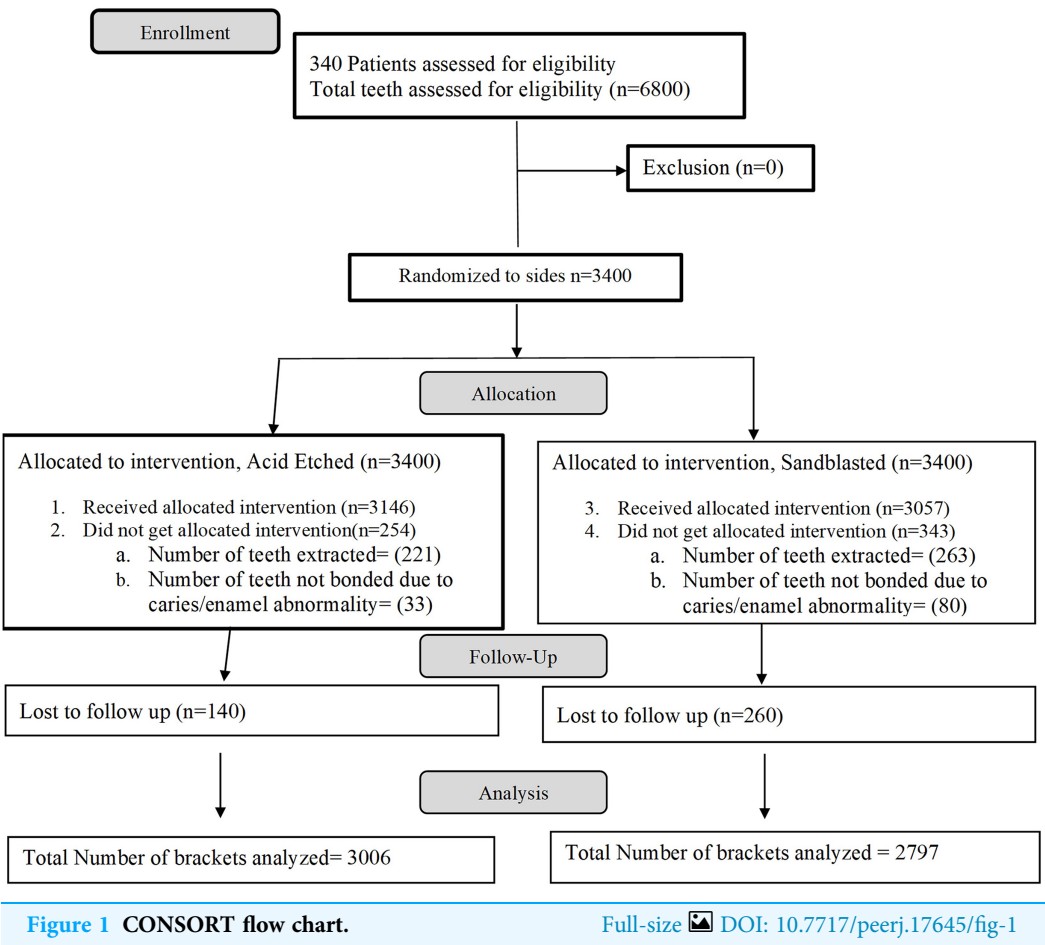

**Figure 1 CONSORT flow chart.**

brush for 1 min, followed by washing with distilled water. The brackets were then air-dried and stored.

The primary outcome of the study was the first-time debonding of the bracket from the tooth after its initial placement, referred to as failure in this study. Patients were advised to visit the orthodontist on a monthly basis for routine orthodontic adjustments, during which any bracket failures were recorded on a data collection sheet. Debonding of the brackets was observed for 18 months after the initial bond. If a debonded bracket, particularly on the lower teeth, was knocked off due to premature contact with the upper teeth, it was not considered for the study. No changes were made to the study design after its commencement.

Sample size calculation was done using the formula described by *Pocock (1983)* and recommended by *Pandis, Polychronopoulou & Eliades (2011)*. Using the failure rate reported by *Ozer & Arici (2005)* of 4.9% failure of sandblasted brackets. The calculation was based on the formula:

$$n = f(\alpha/2, \beta) \times [p1 \times (100 - p1) + p2 \times (100 - p2)] / (p2 - p1)2$$

where $p_1$ is 4.9% (failure of sandblasted brackets), $p_2$ is 4.3% (failure of control group) according to the study of Ozer and Arici, also $\alpha$ is 0.05 and $\beta$ is 0.8. This resulted in a

requirement of 416 brackets per group to have a 95% chance of detecting a significant difference at the 5% level. The sample size in this study was well above the minimum required to account for attrition and patient loss.

In this study, patients who required orthodontic brackets in both the upper and lower arches were included. A total of 5,803 brackets were placed, with each patient receiving both types of brackets in a contralateral split-mouth design. The brackets in the two groups were randomly distributed using Random Allocation Software (version 2.0) in a 1:1 distribution to the left and right sides of the maxilla. The contralateral sides in the mandible received the same type of bracket. The sequence of randomization for patients and allocation of groups to the selected patients were managed by a department assistant who had no financial or research interest in the study. The main outcome measure was the debonding/failure of the bracket.

The operator, the patients, and the data analyst were blinded to the types of brackets bonded to the randomized teeth. Blinding was achieved for the operator by masking the base of the bracket with composite before it was handed to the operator by the dental assistant. Patients were also unaware of the type of bracket they received on either side, as there was no visual difference between the brackets once bonded to the tooth surface. Furthermore, the outcome assessment was blinded. Data were gathered monthly, continuing until the completion of treatment or until the bracket had debonded.

The data were analyzed using SPSS (version 20; IBM, Armonk, NY, USA). Kaplan-Meier survival analysis was performed to evaluate the bracket survival times. Both cumulative survival curves and log survival curves were plotted to visualize the differences between the two groups. A statistical comparison of the groups was conducted using the log-rank (Mantel-Cox) test to determine if there were significant differences in survival distributions. This analysis provided insights into the longevity and failure rates of the different types of brackets under study.

## RESULTS

A total of 310 patients participated in the study, comprising 81 males and 229 females. The acid-etched brackets group included 2,797 brackets, achieving a success rate of 97.3%, with 76 brackets debonding during the 18-month period and a mean survival time of 17.674 months. The sandblasted brackets group consisted of 3,006 brackets, showing a success rate of 97.4%, with 78 brackets debonding over 18 months and an identical mean survival time of 17.674 months (Fig. 2). The survival distribution between the two groups, analyzed using the log-rank (Mantel-Cox) test, was not statistically significant ($\chi^2 = 0.062$, $P = 0.804$). The cumulative survival curves from the Kaplan-Meier analysis were closely approximated, indicating similar bracket failure rates in both groups (Fig. 3). Additionally, scanning electron microscope images taken at a magnification of 2,500x for both types of brackets are presented side by side for comparative analysis (Fig. 4).

## DISCUSSION

Numerous studies have compared the bond strength of sandblasted brackets to non-sandblasted ones when bonded with various adhesives (*Algera et al., 2008*; *Linklater &*

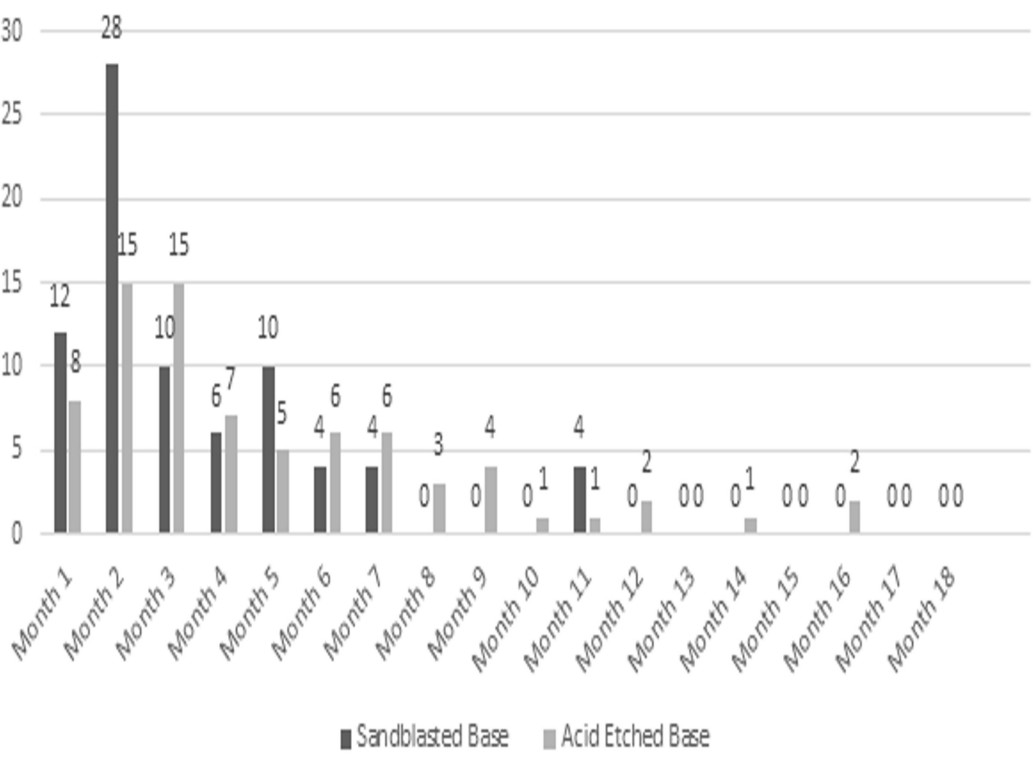

**Figure 2** Month wise failure of sandblasted brackets.

**Log Survival Function**

**Figure 3** Kaplan-Meier plot.

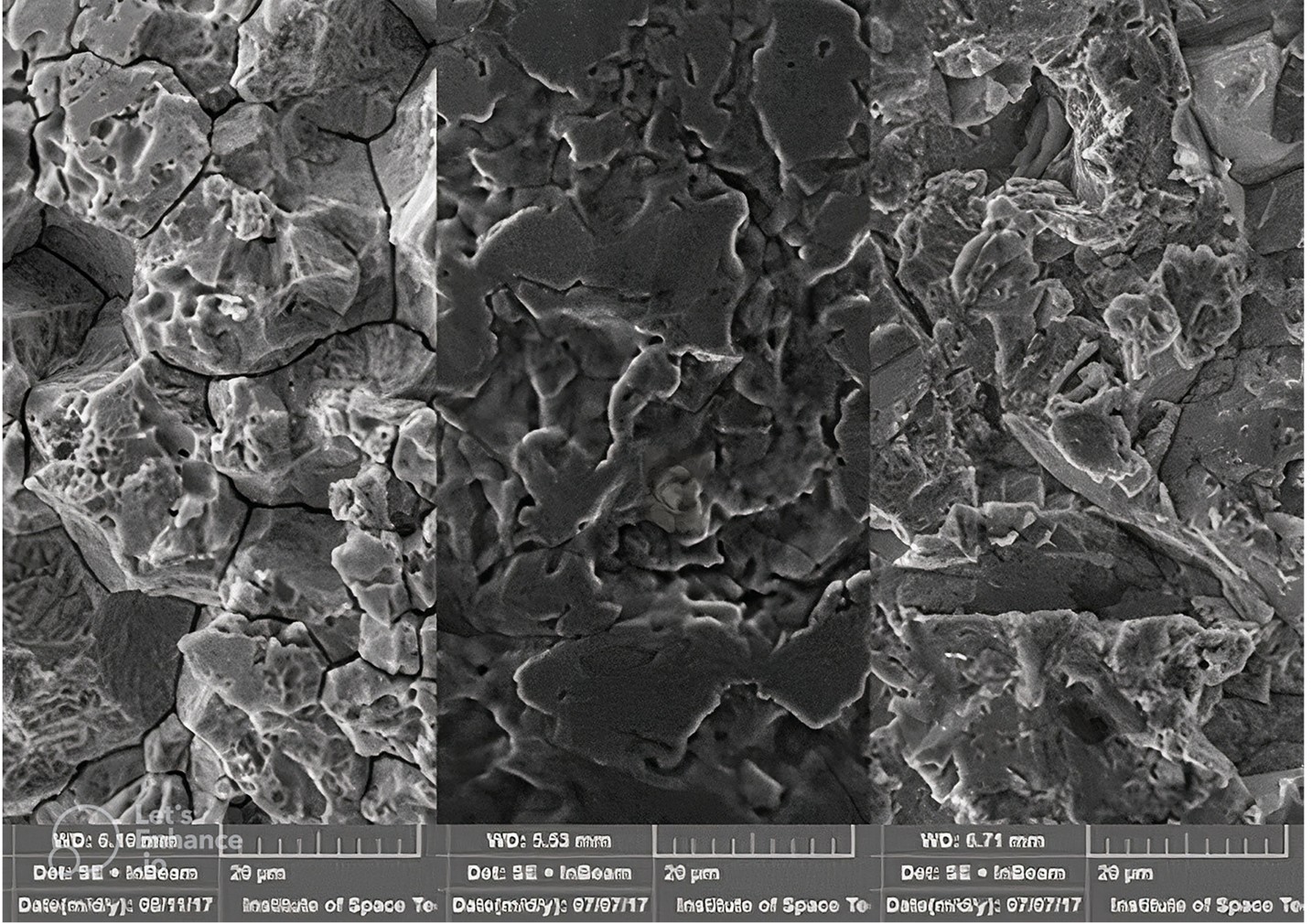

**Figure 4 Scanning electron micrograph of the three brackets used in the study. Left: acid etched; middle: untreated; right: sandblasted.**

*Gordon, 2003*; *MacColl et al., 1998*; *Millett, McCabe & Gordon, 1993*) conducted laboratory studies that concluded sandblasted bases have superior bond strength compared to conventional ones. These findings, along with the understanding that sandblasting increases the surface area and roughness of the base, enhancing the mechanical bonding of composite, have led many manufacturers to incorporate sandblasting in the production of metal brackets.

However, sandblasting brackets is time-consuming and requires specialized equipment. Acid etching with 10% hydrofluoric acid (HF), commonly used to etch ceramic surfaces in clinical setups, presents a simpler, time-saving alternative that does not require special equipment. It is safe for use in concentrations of 20% or less, according to *McKee et al. (2014)* and *Kirkpatrick, Enion & Burd (1995)*. In clinical practice, the etching solution can be directly applied to the bracket base for 60 s and then washed with water. This process

changes the stainless-steel surface from a glossy silver to a dull grey, indicating increased surface roughness.

While it is not recommended to routinely acid etch all brackets, selectively etching brackets with a higher incidence of failure, such as those on premolars and molars, can mitigate the effects of acid exposure to the operator and improve bond strength where it is most needed. Although 10% hydrofluoric acid is routinely used in dentistry to etch porcelain crowns for bracket bonding, it is paramount to use proper precautions when handling corrosive acids. The same method can be applied chairside to the base of orthodontic brackets as needed, mirroring the procedure used for porcelain crowns. Proper personal protective equipment (PPE) such as gloves, safety goggles, and protective clothing should always be worn when handling hydrofluoric acid to prevent skin burns and other injuries. Adequate ventilation and following manufacturer guidelines for acid handling are also essential to ensure safety and effectiveness.

There was no statistical difference in the failure rate between the two groups in our study, suggesting that the roughness achieved with the acid etch can provide similar adhesion to that of sandblasting. To study the physical effects of acid etching and sandblasting a field emission electron microscope (Mira Tescan 3; Tescan, Brno, Czech Republic) was used to acquire magnified images (Fig. 4). It can be ascertained that there is preferential dissolution of metal at the grain boundaries, also pitting of around 10 microns can be seen. It can be appreciated that both methods increase the roughness/surface area for the enhanced interlocking of the composite resin.

*In-vitro* mechanical data cannot be wholesomely applied to an *in-vivo* scenario, as suggested by *Thanos, Munholland & Caputo (1979)* in 1979. Various laboratory studies have employed diverse methods to replicate the oral environment, yet the complexity of intraoral conditions presents a significant challenge. According to Oilo, the oral environment is an intricate amalgamation of numerous simultaneous processes, rendering exact replication outside the mouth exceptionally difficult, if not impossible. *Oilo (1992)* concluded that, to date, no definitive correlation exists between *in-vivo* and *in-vitro* tests. Similarly, *Sunna & Rock (1999)* also stated that there is no correlation between the bond strengths and the clinical failure rate of brackets bonded with composites.

*Murray & Hobson (2003)* suggested that composites likely undergo greater degradation *in vivo* than *in vitro* due to several factors. These include the continuous interaction of forces from mastication and orthodontic wires, the variability of temperatures in the oral environment, the enzymatic degradation of composites by saliva, and the corrosion of brackets. Despite the challenges, most dental research into composites persists in laboratory settings due to the difficulty of exposing materials to the oral environment without altering it or imposing on the compliance of subjects (*Murray & Hobson, 2003*). Given these considerations, our study aimed to evaluate bracket performance in an actual clinical environment, striving to bridge the gap between *in-vitro* findings and *in-vivo* applicability. This approach ensures that the results reflect the complex and dynamic conditions present in the oral cavity.

*Smith & Reynolds (1991)*, as well as *Maijer & Smith (1981)*, conducted *in-vitro* studies on brackets from various manufacturers and concluded that mesh bracket bases provided

the best resin penetration and bond strength (*Din et al., 2022*). Previous research has explored the *in-vitro* bond strength of sandblasted brackets, with findings generally recommending sandblasting to enhance shear bond strength in universal testing machines (*MacColl et al., 1998*). However, there is limited data on the *in-vivo* survivability of sandblasted brackets and a notable lack of data on acid-etched brackets in clinical settings.

The present study reveals a statistically insignificant difference in bracket failure rates between acid-etched bases (2.71%) and sandblasted bases (2.59%). These findings are comparable to the clinical study by *Ozer et al. (2014)* which reported a 2.97% failure rate for brackets with micro-etched bases, predominantly affecting premolar brackets. This suggests that while both sandblasting and acid-etching techniques are effective in improving bracket adhesion, their performance in a clinical environment is similar, offering reliable alternatives for orthodontic treatment. These insights underscore the importance of evaluating bracket bonding methods not just in laboratory settings but also in actual clinical scenarios to ensure their practical effectiveness and longevity.

One of the limitations of this study is that it exclusively examines the effects of sandblasting and acid etching on a single brand of mesh base brackets. Consequently, the generalizability of these findings to other types of stainless-steel brackets remains uncertain. Additionally, the study did not include a control group of brackets that were neither sandblasted nor acid-etched. Incorporating such a control group would have allowed for more comprehensive and relevant comparisons, enhancing the robustness of the outcome assessment. Future research should consider these aspects to broaden the applicability and relevance of the findings across different bracket types and treatment conditions.

## CONCLUSION

This study concludes that acid etching of the bases of orthodontic brackets can be effectively used as an alternative to sandblasting. Acid etching offers a practical advantage as it can be conveniently performed chairside, eliminating the need for specialized equipment required for sandblasting. This method maintains comparable clinical performance in terms of bracket retention and failure rates, suggesting its viability as a time-efficient and accessible option for enhancing bracket adhesion in orthodontic treatments. Future research should explore its applicability across various types of brackets and clinical settings to further validate these findings.

### Funding

The research was funded by Researchers Supporting Project number (RSPD2024R950), King Saud University, Riyadh, Saudi Arabia. The funders had no role in study design, data collection and analysis, decision to publish, or preparation of the manuscript.

## Grant Disclosures

The following grant information was disclosed by the authors:
Researchers Supporting Project number: RSPD2024R950.
King Saud University.

## Competing Interests

The authors declare that they have no competing interests.

## Author Contributions

- Owais Khalid Durrani conceived and designed the experiments, authored or reviewed drafts of the article, and approved the final draft.
- Ulfat Bashir Raja conceived and designed the experiments, authored or reviewed drafts of the article, and approved the final draft.
- Farooq Ahmad Chaudhary conceived and designed the experiments, analyzed the data, prepared figures and/or tables, and approved the final draft.
- Umar Hamid performed the experiments, analyzed the data, prepared figures and/or tables, and approved the final draft.
- Muhammad Qasim Javed performed the experiments, analyzed the data, prepared figures and/or tables, and approved the final draft.
- Sundus Atique performed the experiments, prepared figures and/or tables, authored or reviewed drafts of the article, and approved the final draft.
- Syed Rashid Habib performed the experiments, authored or reviewed drafts of the article, and approved the final draft.

## Human Ethics

The following information was supplied relating to ethical approvals (*i.e.*, approving body and any reference numbers):

The ethical approval was taken from the Ethical Committee of Islamic International Dental Hospital , Riphah International University (IIDC/IRC/06/06/2020).

## Data Availability

The raw measurements are available in the Supplemental Files.

## Clinical Trial Registration

The following information was supplied regarding Clinical Trial registration:

NCT04456114

## Supplemental Information

Supplemental information for this article can be found online at http://dx.doi.org/10.7717/peerj.17645#supplemental-information.

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
