# Peer review of "Comparative in-vivo bond failure rate of orthodontic brackets when bracket base is treated with micro-abrasive blasting vs. acid etching: eighteen month randomized control trial and scanning electron microscope study"

_PeerJ, doi:10.7717/peerj.17645_

## Round 0.1 · original submission · Major Revisions

Dear Authors,

After a thorough review of your paper, I kindly request major revisions to address specific concerns. First, provide clarity on the power calculation by defining all relevant terms. Second, reconsider the recommendation of hydrofluoric acid usage, given its hazardous nature. Third, acknowledge the absence of testing an untreated control bracket as a study limitation.

Figure 3 is important and should be improved instead of removed. Please modify Figure 3's y-axis (Cum Survival) to display only the range from 0.8 to 1.0. In this way, the differences between groups would be enhanced and more easily observed by the reader.

Your attention to these points is crucial, and I eagerly await the revised manuscript.

Best regards,
Section Editor

**Language Note:** The review process has identified that the English language must be improved. PeerJ can provide language editing services - please contact us at [email protected] for pricing (be sure to provide your manuscript number and title). Alternatively, you should make your own arrangements to improve the language quality and provide details in your response letter. – PeerJ Staff

·

Basic reporting

The manuscript was relatively clearly written, with minor deficiencies in formatting appearing throughout the document. For example, the use of "et al." when refering to publications was sometimes used appropriately, and other times not. If there are more than 2 authors, et al. should be used. If there are 2 authors, then both authors should be listed and "et al." would not be used. Also, references are usually referred to within the sentence, before the period, and not listed after the period.

The literature was generally used appropriately to support statements, although there was no justification given for selection of hydrofluoric acid as acid etching agent. This was a notable omission, given that the acid is considered a hazardous material and special safety precautions should be used when handling the acid.

To help clarify the goals of the study, it is typically useful to state a hypothesis, often at the end of the Introduction. Because there were no hypotheses stated, it could not be judged if the results were relevant to a hypothesis.

Experimental design

The research could be appropriate for this journal, issues with the study and write-up not withstanding.

The research question was defined, although if hypotheses were stated, the intent would have been clearer.

The power calculation needs clarification - please define all the terms.

To this reviewer, it is not clear that use of hydrofluoric acid is an appropriate recommendation, given the hazardous nature of the acid. Although the authors state that safety is not a concern if the acid is used in concentrations less than 20%, this information is not consistent with many other recommendations found on-line. For example, some sources state that concentrations as low as 3% can cause burns.

No details were provided on how the hydrofluoric acid was applied, only that it was applied for 1 minute. More information is needed on exactly how it was applied, and if any precautions were taken to avoid accidental exposure.

Validity of the findings

For the Results section, it would have been good to describe what was seen in the electron microscopy images. It was only mentioned that these were obtained and a Figure was shown, but not until the Discussion section were the findings presented.

There was confusion on the timeframe of the study. In some places monitor for 18 months is mentioned, and other places findings at debonding were also described.

It is not clear why if bracket debonds were found at the debonding appointment that these findings were not included in the study. If only bracket debonds for the first 18 months were assessed, then it would be best not to mention looking at debonded brackets at the debonding appointment.

To add validity to the clinical component of this study, a control group should have been used where brackets were untreated. Therefore, ideally there should have been 3 groups of brackets: control/untreated, abrasive blasted treatment group, and acid etched treatment group. Perhaps this deficit could be compensated for if there are other studies that could be dicussed that compared sand blasted brackets to control brackets.

Would be nice to have more discussion on the impact of acid etching. Is there literature on the effects of acid etches on metal materials? How do the findings of other studies compare with yours?

The conclusion that acid etching can be used chairside was not based on findings of this study. My understanding is the brackets were etched beforehand, presumably in a laboratory environment. Given the hazardous nature of the acid, it is likely inappropriate to use the etching material chairside, particularly since alternatives are available.

Additional comments

This was a relatively straight forward study with clinically-applicable results, but the content (writeup) of the manuscript could be notably improved. It is likely that the safety of using 10% hydrofluoric acid should not be played up as the acid is considered a hazardous material at this, and lower concentrations. The manuscript would be enhanced and of better service to clinicians if there was mention of the caution needed when handling hydrofluoric acid.

Reviewer 2 ·

Basic reporting

The manuscript is well written. The English language is professionally used throughout the whole manuscript. The literature is well documented, and previous work has been cited. The structure of the paper is good, and the flow of headings and subheadings is OK.

Experimental design

The aims are well illustrated.
The methodology is sound.
The investigation has been well documented.
Sufficient details are given for each step.
The study has no ethical concerns.
The follow-up period is good, and the study design is appropriate.

Validity of the findings

The study presents the efficacy of using hydrofluoric acid in preparing bracket bases before bonding compared to sandblasting. The results showed the possibility of using this technique to improve the survival rate of bonded orthodontic practice. I think that the statistical analysis is satisfactory. The presented results are fine.

Additional comments

No comment

Reviewer 3 ·

Basic reporting

• Some of the language/grammar throughout should be adjusted for clarity and adherence to the standards of English. It is suggested that the authors employ an English editing/proof-reading service
• While working to improve the bond strength and survival of orthodontic brackets is admirable, the authors should provide additional introduction to explain why this improvement over the stock brackets was necessary. As the stock brackets used already had mesh backing.
• It is suggested that the authors use a different font for figures 3 and 4, because the text is difficult to read.

Experimental design

• The lack of testing an untreated control bracket is concerning, and at the very least it should be included as a limitation of the study.
• It would be good for rigor if the authors performed an SEM x-ray map of the surface of some of the treated and untreated bracket bases, because the authors said in their methods section (123, 124, 125) that the ultrasonic cleaning removed residual silica particles that may have been embedded. The SEM x-ray map of the surface would confirm if this cleaning was successful.
• There appear to be inconsistencies in the number of patients reported: line 59 states 340 patients bonded; line 99 states 310 patients were recruited; line 171 states 310 patients participated; page 1 of the consent form states that up to 316 patients will participate. This may just be a typographical error but it should be corrected.
• The failure types of the bonding should be mentioned (for example: mixed, adhesive failure between bracket and composite, adhesive failure between composite and enamel). This is important because too high of a bonding strength can cause damage to the enamel during debonding. Pictures of representative failures should be included.
• Page 1 of the consent form, under the heading “Study Procedures” states that either “conventional acid etch technique or using sandblast technique” will be applied, which is inconsistent with the earlier paragraph on the same page (“Purpose of the Study”) stating “compare the survivability of orthodontic brackets with a chemical etched base versus a sandblasted base bonded with Transbond XT using the conventional acid etch technique.” This could be very confusing to any participants who are not familiar with orthodontics and unaware that the conventional acid etch technique refers to phosphoric acid etching and not hydrofluoric acid etching; and potentially misleading to communicating the purpose of the study to any potential participants.
• What were the modifications to the study protocol alluded to in lines 100, 101, 102?

Validity of the findings

• Kudos to the authors for restricting their conclusions to only comparing the bonding survival of acid etched and sandblasted brackets to each other. However, the very close survival percentage and presented statistics of both experimental groups causes me to wonder if the sandblasting and HF etching was even necessary; an untreated control would make this clearer. That said, it is good to see brackets evaluated in clinical use because there are severe limitations to in-vitro testing.

Additional comments

• What magnification was used for the SEM images?
• Of particular concern is the statement in line 83, 84, 85 that “Hydrofluoric acid does not cause any damage to human skin at concentrations below 20%”, because it is contradicted by the references provided by the authors. Concentrations of HF below 20%, and as low as 1%-3%, can definitely cause burns. Further, skin exposure to lower concentrations can simply delay symptoms and effects of more serious complications. So, when working with HF it is always important to have the Calgonate Calcium Gluconate cream first aid kit available, and to use appropriate gloves. As well as being aware that HF is a dangerous chemical.

Reviewer 4 ·

Basic reporting

The manuscript is clear written, but with some errors.

The title of the manuscript must be rephrased, it's to long.
In the line 188 the sentence must be corrected.
Figure 3 is not concluding. I suggest to be changed.
References must be updated according to current research, most of them are too old.

Experimental design

In the manuscript is clear presented the aims of the research, in accordance with requirements of the journal.
Methods is described with sufficient detail and information.

Validity of the findings

Statistical analysis I suggest to be more complex.

Conclusions are well stated.

Additional comments

The research approach in this manuscript is interesting, but for the publication is necessary to be improved.

---

## Round 0.2 · Major Revisions

Dear author,

Thank you for submitting your revised manuscript titled "Comparative in-vivo bond failure rate of orthodontic brackets when bracket base is treated with micro-abrasive blasting vs acid etching: eighteen month randomized control trial and scanning electron microscope study" to PeerJ. While I appreciate your efforts in addressing the reviewers' comments, I regret to inform you that I am unable to accept the manuscript in the current stage and therefore recommend new major revisions before further consideration for publication.

The reviewers have raised significant concerns regarding the manuscript, including issues with grammar, experimental design, validity of findings, and organization. It is crucial that you carefully address all comments from both rounds of review to ensure the manuscript meets our publication standards.

I encourage you to thoroughly revise the manuscript, paying particular attention to clarity, organization, and accuracy of discussion. Once the revisions are complete, please submit the revised manuscript along with a detailed response letter outlining the changes made.

Thank you for your attention to these matters. I look forward to receiving your revised manuscript.

Sincerely,
Dr. Tribst JPM

**Language Note:** The Academic Editor has identified that the English language must be improved. PeerJ can provide language editing services - please contact us at [email protected] for pricing (be sure to provide your manuscript number and title). Alternatively, you should make your own arrangements to improve the language quality and provide details in your response letter. – PeerJ Staff

·

Basic reporting

Although overall the content is improved over the first submission of this manuscript, the need for notable improvements remain, particularly regarding to the quality of the writing. The Introduction, for example, is one long paragraph. The contents would be easier to follow if the information was broken up by paragraphs, each containing information on particular topics. The same situation appears in all sections of the manuscript, where information could be presented more clearly if more paragraphs were used.

Overall, ideally, it would be good to have someone proofread the manuscript who is fluent in English and knowledgeable of the topic.

Experimental design

The inclusion and exclusion criteria for subjects should be consolidated and presented in a unified manner. Currently the description of the criteria is interrupted by the inclusion of information on how the brackets were bonded, or other information, and elements of the criteria are interspersed throughout the Materials and Methods section. Perhaps it would be helpful to construct an outline of topics that are to be presented, and use this to guide to help improve the organization of this section.

As previously mentioned, the study lacks a control group (brackets that were not treated either by sand blasting or acid etching), and this is a notable omission in the study design, even if this has now been acknowledged as a limitation.

Validity of the findings

The discussion section is weak and needs major reorganization. The first half discusses the weaknesses of in vitro studies that were performed by others, with no reference to the findings of the current study. Discussion of the results of the current study first appear a page and a half into this section, whereas typically discussion of the results should begin at the outset of this section. My suggestion would be to reorganize the sequence of information presented in this study. The weaknesses of in vitro studies could be mentioned after the results of the current study are directly discussed, and before the limitations of the study are mentioned.

Additional comments

It is unclear how the acid etch method could be used chairside, as is mentioned, due to the need for rinsing of the brackets before they are bonded. Etching the brackets chairside would be cumbersome whereas the method would seemingly be more ideally suited as a lab procedure, as was done for the current study.

Given the extent of work that remains to improve this revised manuscript, unfortunately my enthusiasm on prospects for publication has diminished.

Reviewer 2 ·

Basic reporting

Everything is fine.
I have already given my positive comments in my previous review of this manuscript.

Experimental design

Everything is fine.
I have already given my positive comments in my previous review of this manuscript.

Validity of the findings

Everything is fine.
I have already given my positive comments in my previous review of this manuscript.

Additional comments

Everything is fine.
I have already given my positive comments in my previous review of this manuscript.

Reviewer 3 ·

Basic reporting

There are some minor issues with English grammar, nothing that significant, just a few spots here and there where the grammar could be fixed.

Experimental design

Were the silicon dioxide particles beads (ie rounded) or shards (ie sharp like grains of sand)?

Validity of the findings

In addition to their initial sample size calculation (which suggested a size of 416 brackets per group) did the authors calculate what effect size could be detected with their final sample size of 5,803?

Additional comments

In lines 81-83, the authors should clarify that the studies by MacColl and Sharma et al were conducted on metal brackets. Because while sandblasting does generally increase the bond strength of metallic brackets, it is deleterious for ceramic ones, and is specifically recommended against by the manufacturer. See "Shear Bond Strength and Bracket Base Morphology of New and Rebonded Orthodontic Ceramic Brackets" by Mihai Urichianu et al as an enxmple.

The statement on lines 87-88 that Hydrofluoric acid below 20% concentration does not damage human skin is misleading at best and incorrect at worst; even by their cited references:

"A review of hydrofluoric acid burn management" by McKee et al. 2014
"Pain out of proportion to physical examination is a hallmark finding in HF burns. Clinically, the morbidity of the burn is directly proportional to the concentration of HF, the duration of exposure, and the immediacy and adequacy of first aid measures (eg, copious irrigation). In the industrial setting, concentrations can reach levels >20%; however, the majority of patients are burned at 1% to 3% concentration, more commonly present in cleaning solutions and solvents (4–6). It is useful to categorize exposures based on the concentration of acid. Higher concentrations of acid results in more immediate pain and visible burn, followed by the development of grey areas, necrosis or ulceration, and possibly tenosynovitis and osteolysis, which can present as late manifestations. Lower concentrations (<20%) could result in delayed symptoms up to 24 h postexposure and, if left untreated, could progress through the same sequence as the high-concentration burns."

HF is commonly used and is a useful acid, but its safety cannot be minimized because it is a dangerous acid.

---

## Round 0.3 · accepted · Accept

Dear author,

Thank you for submitting your revised manuscript titled “Comparative In-Vivo Bond Failure Rate of Orthodontic Brackets When Bracket Base Is Treated with Micro-Abrasive Blasting vs. Acid Etching: An Eighteen-Month Randomized Control Trial and Scanning Electron Microscope Study” to PeerJ. I appreciate your efforts in addressing the reviewers’ comments. You have diligently addressed all of the reviewers’ feedback, resulting in substantial improvements to the manuscript. I am satisfied with the current state of the manuscript.

Based on the revisions made, I am pleased to confirm that the manuscript is now ready for publication.

Best regards,
Dr. Tribst JPM

·

Basic reporting

Much improved over the original submission.

Experimental design

The design is clearly described.

Validity of the findings

All looks good with the presentation and analysis of findings.

Additional comments

The work of the authors is appreciated. The latest version of the manuscript reads well!